# The Effects of Varying Heat Treatments on Lipid Composition during Pelagic Fishmeal Production

**Gudrun Svana Hilmarsdottir [1],\*** , **Ólafur Ogmundarson [1]** , **Sigurjón Arason [1,2]** and **María Gudjónsdóttir [1,2]**

[1]  Faculty of Food Science and Nutrition, University of Iceland; Aragata 14, 102 Reykjavík, Iceland; olafuro@hi.is (Ó.O.); sigurjar@hi.is (S.A.); mariagu@hi.is (M.G.)
[2]  Matís ohf, Icelandic Food and Biotech R&D, Vínlandsleið 12, 113 Reykjavík, Iceland
\*  Correspondence: gsh9@hi.is

**Abstract:** The study aimed to provide insight into the lipid quality of pelagic fishmeal and fish oil processing of mackerel and herring cut-offs, and the effect of temperature changes in the cooker (85–95 °C) during production. Samples were collected after each processing step at a traditional processing line where water and lipid content, free fatty acids (FFA), phospholipids (PL) and fatty acid composition (FAC) were measured. Results showed that the standard procedures at 90 °C included ineffective draining and concentration steps. Moreover, the solid streams entering the driers variated in chemical composition, suggesting that processing each stream separately could be beneficial for maintaining the lipid quality. The cooking temperature affected the lipid removal from the fishmeal processing, where lowering the temperature to 85 °C resulted in a lower lipid content of the final fishmeal, along with lower FFA and PL values. Hence, the fishmeal and fish oil factories could save energy by lowering the cooking temperature, as well as obtaining more stable and higher value products. Further recommendations include more focus on the initial steps for a better homogenization and breakdown of the raw material, as well as investigation of different drying techniques applied on each processing stream entering the drying steps.

**Keywords:** fishmeal; fish oil; process optimization; heat treatment

## 1. Introduction

Marine rest raw materials (including remains from main production lines, cut-offs, heads, guts, by-catch, etc.) are a great source of lipids, proteins and minerals and have been used in fishmeal and fish oil production, along with small pelagic species [1]. Fishmeal and fish oil are considered the most nutritious and digestible ingredients for farmed fish, and with no major increases in raw material, any increase in fishmeal production needs to come from byproducts [2]. The current estimate of cut-offs from the main production that enter the fishmeal and fish oil factories is 25–35% of the total volume [2]. Although it is a positive development, the traditional fishmeal and fish oil processes were developed in the 1940s to 1960s [3], and little improvements have been made to the land-based processes since then [1,4–7]. However, applied handling improvements from catch to landing have resulted in higher fishmeal quality [8]. Hence, most of the produced fishmeal remains with a high lipid content, which the Food and Agriculture Organization (FAO) of the United Nations defines as a Type C fish protein concentrate (FPC), or fishmeal processed under sufficient hygienic conditions. Type C FPCs contain rancid lipids that can lower the nutritive value of the proteins, affect the product flavor and odor, and increase the risk of cumulative toxic effects if consumed regularly over a long period [1,9]. However, if the lipid content of the FPC is lowered below 0.75 g/100 g sample, the highest FPC class (Type A) would be reached, allowing improved FPC for human consumption [1,9]. Further

regulations for fishmeal and fish oil come from the Marine Ingredient Organization (IFFO), which is guided by the FAO regulations and accounts for more than 75% of the fishmeal and fish oil trade worldwide [10].

Heating is one of the most critical processing steps in fishmeal and fish oil production, both during cooking and drying. Cooking is the main step intended to separate the lipids and the proteins, making the lipid extractions more efficient at later stages of the processing line. To keep the lipid content low in the final fishmeal, it is important to remove the majority of the lipids early in the process, as the lipids cannot be extracted during drying. Therefore, by cooking the raw material at a temperature where the separation of the lipids and the proteins is the most effective, the lipid content in the fishmeal could be lowered. Moreover, above 90 °C, intermolecular disulfide bonds start to form and protein coagulation thereafter [11], leaving unfavorable interactions with solvent water [12]. Furthermore, cod and herring muscle proteins deform at different temperatures, starting from approximately 30 °C, while most of the proteins were fully unfolded at approximately 90 °C [13]. These results indicate that muscle protein degradation and denaturation is highly dependent on the chosen heat treatment [14].

Iceland's most caught pelagic species are the Atlantic herring (*Clupea harengus*) and capelin (*Mallotus villosus*). However, in recent years, catchings of oceanic redfish (*Sebastes mentella*), blue whiting (*Micromesistius poutassou*), and Atlantic mackerel (*Scomber* scombrus) have increased [15,16]. When the mackerel is caught in Icelandic waters, it has generally been feeding on the zooplankton species *Calanus finmarchicus* [17,18]. *C. finmarchicus* is very rich in enzymes, which can have fast degradative effects on the landed mackerel raw material if not treated properly. Hence, the processing companies tend to behead and gut the mackerel to prolong mackerel shelf life. The heads and guts are collected for fishmeal production along with bycatch and other potentially remaining raw materials. Since it takes a long time to collect the appropriate amount of these side streams prior to process initiation, the raw material must wait several days in the tanks until they are full, and enough material has been collected to initiate the process. This delay prior to operation increases the risk of degradation of the material due to microbial, enzymatic and oxidative processes [19,20]. Moreover, the raw material is highly heterogeneous, as it includes bycatch, heads, viscera, stomach content and damaged whole fish, which is all blended and collected over time. Moreover, as the mackerel catching season overlaps with the herring season, fishmeal is often produced from a mixed catch, i.e., including multiple species, increasing the heterogeneity of the raw material even further as the production pace of fishmeal is too high for separating the catch.

In this context, the aim of this study was to make a detailed investigation of the fishmeal and fish oil production processes of highly diverse and fat raw material from Atlantic herring and Atlantic mackerel and to assess the effects of three different temperatures in the cooker (85 °C, 90 °C and 95 °C) on the final fishmeal and fish oil quality.

## 2. Materials and Methods

### 2.1. Raw Materials and Sampling

Fish were caught east and southeast of the Icelandic coast by purse seiners from September 3 to September 6, 2017. The raw materials entering the fishmeal production line weighed in total 885 tons, and consisted of 513 tons (58%) of Atlantic mackerel cut-offs (*Scomber scombrus*), 330 tons (37%) of Atlantic herring cut-offs (*Clupea harengus*) and 40 tons (4.5%) of blue whiting (*Micromesistius poutassou*). Initiation of the fishmeal processing line was three days post catch as the mackerel and herring blend mainly consisted of cut-offs and damaged fish. The production capacity of the factory is around 10 tons per hour of fishmeal, with 1200 tons of raw material entering the production line per day. During September 2017, the average fishmeal production yield was 22.5%, and the fish oil production yield was 17.0%.

An overview of the fishmeal and oil process can be seen in Figure 1. Upon initiation of the fishmeal and fish oil process, the raw material entered a **pre-heating** step, where the temperature was kept at

approximately 55 °C for 20 min, followed by a *cooking step at 85–95 °C* for 20 min. The pre-heating step is powered by excess steam or condensate from the evaporators and other equipment for better energy efficiency, lowering the energy cost in the fishmeal plant [1]. Next, the raw material was drained before the press to remove excess water. The press liquid was combined with the drained liquid, which both entered a decanter. These liquid streams combined are called the *separated press liquid*, which was treated both with centrifuges and evaporators to separate the fish oil from the solid streams. Next, a large part of the water was evaporated in a vaporizer before the material entered the drying steps. The solid streams from the press (*press cake*) and the decanter (*sludge*) were combined with the latter *concentrate* in a two-step drying process. The first drying step consisted of a rotary disc steam dryer (steam temperature 160 °C, drying temperature 95 °C and duration time 30 ± 5 min). The second drying step was a Hetland air dryer (maximum input air temperature 450 °C, dryer temperature 150 °C at the middle of the dryer, wet bulb temperature of approximately 65 °C and the drying time was 16 ± 2 min). The steam drying decreased the moisture content of the solid streams to approximately 40–50%, while the air dryer reduced the moisture further to approximately 5–10%. Some fine particle meal (*fine meal*) swirled up in the air duct during the air drying. This meal was lighter than the fishmeal and was collected and blended with the rest of the dried meal to make the final *fishmeal* product.

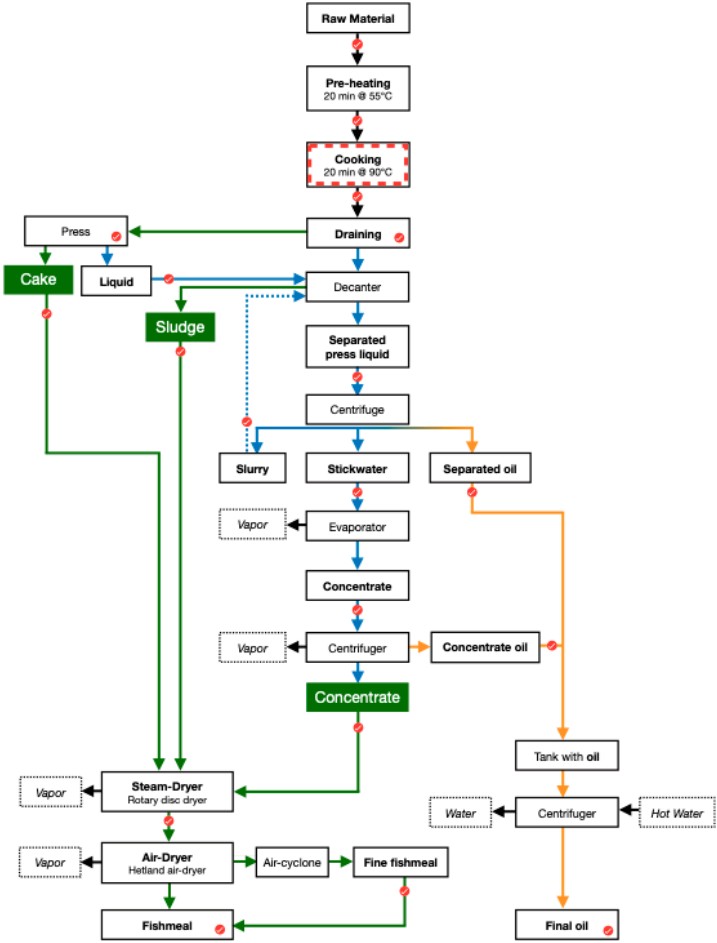

**Figure 1.** A traditional fishmeal and fish oil processing line. The green color represents the solid streams throughout the processing line, the blue streams identify the liquid streams and the yellow streams represent the oil streams. Red dots indicate sampling points in the production. Green-filled boxes highlight the solid streams entering the drying steps, and a red dashed line highlights the cooking step, which was investigated at three different temperatures.

After a steady state process had been established to produce commercial fishmeal, samples were collected throughout the process, with fishmeal and fish oil as end products. Standard cooking conditions include cooking the raw material at 90 °C for 20 min. However, upon changing the cooking temperatures between 85 °C, 90 °C and 95 °C, samples were collected to investigate the effect of the cooking temperature throughout the production line. All samples were cooled to 0 °C ± 2 °C overnight and transported the following morning to the laboratory, where the samples were kept at −25 °C until analysis, which took up to 6 months. Prior to analysis, samples were thawed at 0–4 °C for 12 h or up to 36 h, depending on the water content and the sample size. Three individual samples (triplicates) were collected at each point to investigate if the production was homogenous. Each triplicate was measured twice to confirm the consistency and reproducibility of the measurements.

### 2.2. Chemical Analysis

The water content of the samples, except the oil samples, was measured according to ISO 6496 [21]. Water content in the oil samples was measured using calorimetric titration, performed by an 851 Titrando (Metrohm, Herisau, Switzerland). Total lipids (TL)* were extracted and measured [22], and the TL extracts used both to measure enzymatic lipid hydrolysis in the form of free fatty acids (FFA) [23] with modifications [24] and phospholipid (PL) content [25]. In the current study, PL measurements refer to measurements of phosphatidylcholine, as it is the most abundant phospholipid class in the membrane [26]. The fatty acid composition (FAC) of the samples was determined by gas chromatography (Varian 3900 GC, Varian, Inc., Walnut Creek, CA, USA) of fatty acid methyl esters, based on the AOCS Official Method Ce 1b-89 [27], with minor adjustments. Results for water and lipid content, FFA and PL are shown as g/100 g sample. FAC results are presented as g/100 g lipid.

### 2.3. Statistical Analysis

Data summaries, tables and statistical analyses were performed in Microsoft Office 365 with Excel (Microsoft, Redmond, WA, USA) while one-way ANOVA, Tukey's HSD test and Pearson's correlation were done in RStudio (RStudio Inc., Boston, MA, USA). The significance level was set to $p < 0.05$ for all statistical analyses, and results were shown as mean ± SD from the three triplicates for each sample.

## 3. Results and Discussion

### 3.1. Chemical Composition of Mackerel and Herring Blend at Standard Conditions (90 °C)

3.1.1. Water Content Changes during Standard Processing (90 °C)

The water content increased significantly when the raw material (64.6 ± 2.0 g/100 g sample) was pre-heated (74.6 ± 1.8 g/100 g sample) and cooked (72.9 ± 1.0 g/100 g sample) (Figure 2). This increase in water content can be explained by heterogeneous raw material, as the mackerel and herring blend included different-sized mackerel and herring, in addition to mackerel heads and guts. Due to these large variations in the raw material during the initial processing steps, it can be asserted that during processing of multiple side streams and species, the processing line cannot produce a homogenous blend of raw materials during pre-heating and cooking. No additional solvents or liquids of any kind were added during the fishmeal process. After cooking, the material was drained to separate the heated raw material into solid streams (Figure 1, green-colored dashed line) and liquid streams (Figure 1, blue-colored dashed line).

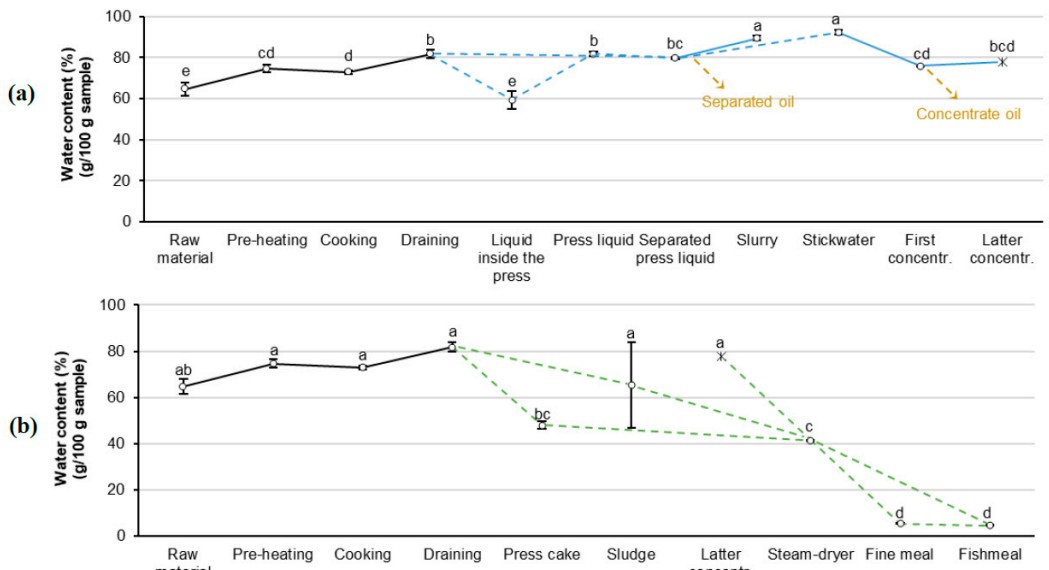

**Figure 2.** Water content in liquid streams (**a**) and solid streams (**b**) from the traditional fishmeal and fish oil processing facilities shown in Figure 1. Liquid streams are identified by a blue color, solid streams by a green color (**b**) and oil streams by yellow color. A dashed line indicates where the process breaks up into multiple streams or where they join each other again. Solid lines indicate only one possible gateway. Letters indicate significant differences where $p < 0.05$.

The draining did not result in a reduction of water content compared with the following step (separated press liquid) of the liquid stream (Figure 2a), indicating the inefficiency of the draining step. However, after pressing, a significant decrease in water content from $81.8 \pm 2.0$ g/100 g sample to $47.9 \pm 1.6$ g/100 g sample was observed, showing effective water removal in the press (Figure 2b). Any remaining solids in the liquid stream were removed in the decanter (sludge), or by centrifugation (slurry). The slurry was recirculated to the decanter because of its high lipid content and large particles. Meanwhile, the stickwater continued throughout the evaporation steps to form the first and second concentrates (Figure 2a). The second concentrate, the sludge (from decanter) and the press cake were then joined in the solid stream and entered the steam dryer. After the steam dryer, the water content had been lowered to $41.5 \pm 0.1$ g/100 g sample, followed by further drying in an air dryer, resulting in a water content of $4.6 \pm 0.2$ g/100 g sample in the final fishmeal (Figure 2b).

Interestingly, the fine meal (that swirls up to the drying cylinder) had a significantly higher water content ($5.4 \pm 0.1$ g/100 g sample) than the final fishmeal. The process overview of the water changes indicated that the press and the air dryer were highly effective in removing water from the fishmeal. However, the draining, concentration and evaporation and steam drying steps were ineffective and required optimization.

During the oil extraction process, water was extracted from the oil by centrifugation, resulting in a final oil with a purity of 99.7 g lipid/100 g sample. As microorganisms and enzymes are primarily active in the water phases of biological samples [28,29], a higher water content would increase the risk of oxidation or other degradation of the oil during storage.

3.1.2. Lipid Content during Standard Processing (90 °C)

The raw material had a lipid content of $19.5 \pm 2.0$ g/100 g sample, which is an intermediate lipid content for mackerel (Figure 3). Mackerel are known to vary in lipid content between catching times, seasons and locations, with an average range between 15–25% [30,31]. In comparison, herring has a lipid content of 5–8% in the white muscle and a 15–20% lipid content in the dark muscle [32]. Herring heads, frames and viscera commonly have a 9–12% lipid content [33]. A negative correlation (r = −0.66) was observed between the water and lipid contents of samples over the whole processing line ($p < 0.05$),

which is in agreement with earlier findings both in mackerel [30] and herring [32]. Thus, the significant water content increase observed during these first processing steps was mirrored by a decrease in lipid content. Approximately 59% of the analyzed lipids in the raw material were removed during the cooking and draining steps, indicating that these processing steps serve an important role in overall lipid separation and removal from the solid stream into the liquid stream. However, the separation might become more effective with the use of other cooking temperatures or other appropriate settings, which will be analyzed in Section 3.2.

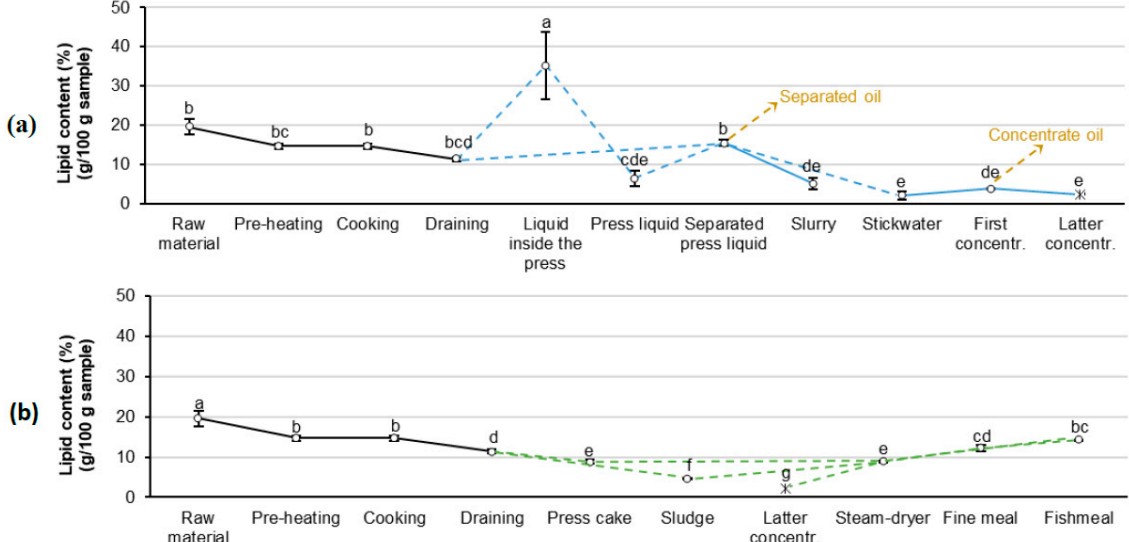

**Figure 3.** Lipid content in liquid streams (**a**) and solid streams (**b**) from the traditional fishmeal and fish oil processing line in Figure 1. Liquid streams are identified by a blue color, solid streams by a green color (**b**) and oil streams by a yellow color. A dashed line indicates where the process breaks up into multiple streams or where they join each other again. Solid lines indicate only one possible gateway. Letters indicate significant differences where $p < 0.05$.

Minor changes in lipid content in the liquid stream (Figure 3a) between the draining and separated press liquid suggest that the small amount of lipid added from within the press (lipid content of $35.2 \pm 8.6$ g/100 g sample) only had a minimal effect on the overall lipid content of the separated press liquid ($15.5 \pm 0.9$ g/100 g sample). The separated press liquid was centrifuged and the oil extracted (the separated press oil), hence decreasing the lipid content in both the slurry and the stickwater. The first concentrate ($3.9 \pm 0.1$ g lipid/100 g sample) underwent a similar oil extraction in addition to evaporation, resulting in a lower lipid content in the second concentrate ($2.4 \pm 0.2$ g/100 g sample).

An overall decrease was observed in the lipid content of the solid stream (Figure 3b) throughout the processing line, until reaching the drying steps ($8.9 \pm 0.2$ g lipid/100 g sample). However, after drying, the relative lipid content in the fishmeal increased to $14.3 \pm 0.3$ g/100 g sample, mainly due to water removal. A significant difference between the fishmeal and the fine meal ($12.3 \pm 0.8$ g/100 g sample) was observed in both the water and lipid contents. The studied fishmeal is considered to be Type C, as it contained a lipid content above 3 g/100 g sample [5,9]. However, high-lipid fishmeal is not uncommon and has been reported before [34], but is hence unsuitable for human consumption, according to the FAO [1,9].

### 3.1.3. Free Fatty Acids (FFA) during Standard Processing (90 °C)

Although up to three days had passed from the catching of the fish until the fishmeal and oil production was initiated, a relatively low free fatty acid (FFA) content (0.4–0.6 g FFA/100 g sample) was observed in the raw material compared with earlier studies [20,32], indicating that enzymatic degradation of the raw material was not severe upon processing. Measured FFA values (Figure 4) were

close to the reported values of a dark herring muscle [32] and, taking into account easier exposure to oxygen for blended cut-offs compared with a part of a muscle, the FFA values are considered relatively low. Keeping FFA values low when catching mackerel can be difficult as its stomach is often full of the enzyme-rich zooplankton *Calanus finmarchicus* [6,17]. Along with gastric enzymes, these enzymes can initiate severe raw material degradation if not treated properly [6,17].

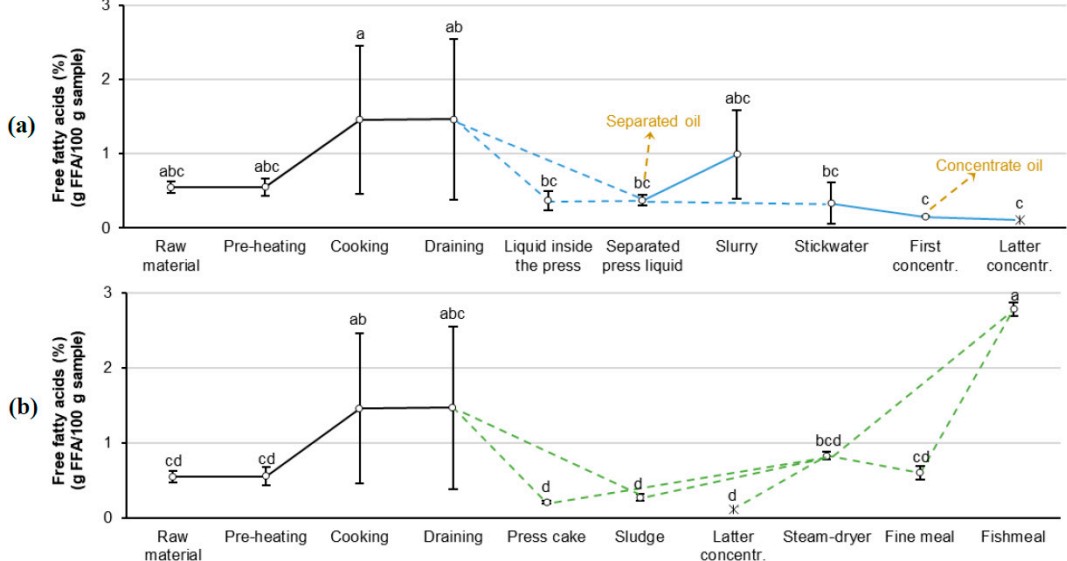

**Figure 4.** Free fatty acid (FFA) content in liquid streams (**a**) and solid streams (**b**) from the traditional fishmeal and fish oil processing line in Figure 1. Liquid streams are identified by a blue color, solid streams by a green color (**b**) and oil streams by a yellow color. A dashed line indicates where the process breaks up into multiple streams or where they join each other again. Solid lines indicate only one possible gateway. Letters indicate significant differences where $p < 0.05$.

The raw materials studied had higher FFA values than fresh mackerel muscle [31], including the effects of seasonal changes. The different parts of mackerel have been reported to have <0.7, <0.5 and <0.4 g FFA/100 g lipid in dark muscle, light muscle and whole mackerel, respectively, if stored at 4 °C for 4 days [20]. The dark muscle of herring has been shown to contain ~2.0 g FFA/100 g lipid and the light muscle ~1.1 g FFA/100 g lipid [32]. The FFA content in the raw material was 2.8 ± 0.7 g FFA/100 g lipid, or 0.5 ± 0.1 g FFA/100 g sample. This implies that the delay before processing enables the activation of enzymes and microbiological spoilage, which induces the formation of FFA through degradation of the raw material and should be kept as short as possible.

Free fatty acids (FFA) were also measured in the liquid, solid and oil streams during processing (Figure 4). Most of the FFA were, however, observed in the solid stream samples, peaking in the final fishmeal (19.3 ± 0.4 g FFA/100 g lipid).

The large standard deviations in FFA during cooking and draining can be explained by the heterogeneous nature of the raw material and the (yet) unoptimized production line. However, the FFA content remained relatively low in the press cake, sludge and latter concentrate, which all contained less than 0.35 g FFA/100 g sample. FFA levels were expected to rise during the concentration of aqueous materials, as observed during drying of the solid streams, but not between the concentration steps [35]. During the air drying, the water content decreased from 41.5 ± 0.1 g/100 g sample down to 4.6 ± 0.2 g/100 g sample, while the lipid content became a proportionally larger part of the sample. Hence, it is not surprising to observe an increase in FFA formation during drying, due to the high temperatures [36] and the raw materials' exposure to oxygen [37]. The FFA content in the fine meal and the fishmeal were 4.3 ± 0.2 g/100 g lipid and 19.3 ± 0.4 g/100 g lipid, respectively. Hence, the lipids of the fine meal were less hydrolyzed or denatured compared with the fishmeal, although the fine meal was lower in water content.

Published results from anchovy meal, processed directly after landing, showed an FFA content of 6.8 g FFA/100 g lipid [38], which is lower than the commercial fishmeal obtained in this study (19.3 ± 0.4 g FFA/100 g lipid). Hence, it is suggested that the processing of fishmeal and fish oil should not wait three days as in the current study. Lower FFA values could be reached by storing the raw material at 2–3 °C or even lower during the wait, or by removing the dark muscle and viscera from the raw material [39].

Moreover, the hydrolysis of PL seems to be a contributor to the increase in FFA if the raw material is not heated and the lipid hydrolysis inactivated [39]. It is reported that the primary cause for FFA escalation in fish oils is contamination by bacteria (genus *Alcaligenes*) which thrives at the oil–water interface, located at the bottom of oil storage tanks, and converts the phospholipids into oil-soluble FFA and water-soluble phosphate esters [35]. As the oil-soluble FFA have a lower density than the oil, they disperse upwards and hence contaminate the oil [35]. In the present study, the final oil contained 0.06 g FFA/100 g lipid, which is within acceptable margins of FFA values of fish oil intended for human consumption [1]. However, the oil would still need to undergo an additional refining process composed of deacidification, transesterification, concentration, deodorization and earth treatment, antioxidant addition and fill off to be considered for human consumption [1,40], in addition to FFA levels below 0.1 g FFA/100 g lipid [1].

### 3.1.4. Phospholipids (PL) during Standard Processing (90 °C)

Phospholipids (mostly phosphatidylcholine [26]) were measured throughout the processing line (Figure 5). Phospholipid values in the raw material were 0.9 ± 0.2 g PL/100g lipid, which is in good agreement with PL values in other pelagic species, as earlier studies have reported PL values ranging between 1.5–3.0 g PL/100 g lipid in herring and 0.7–4.0 g PL/100 g lipid in mackerel [31,32]. Upon cooking and draining, PL were almost non-existent in the liquid streams (Figure 5a).

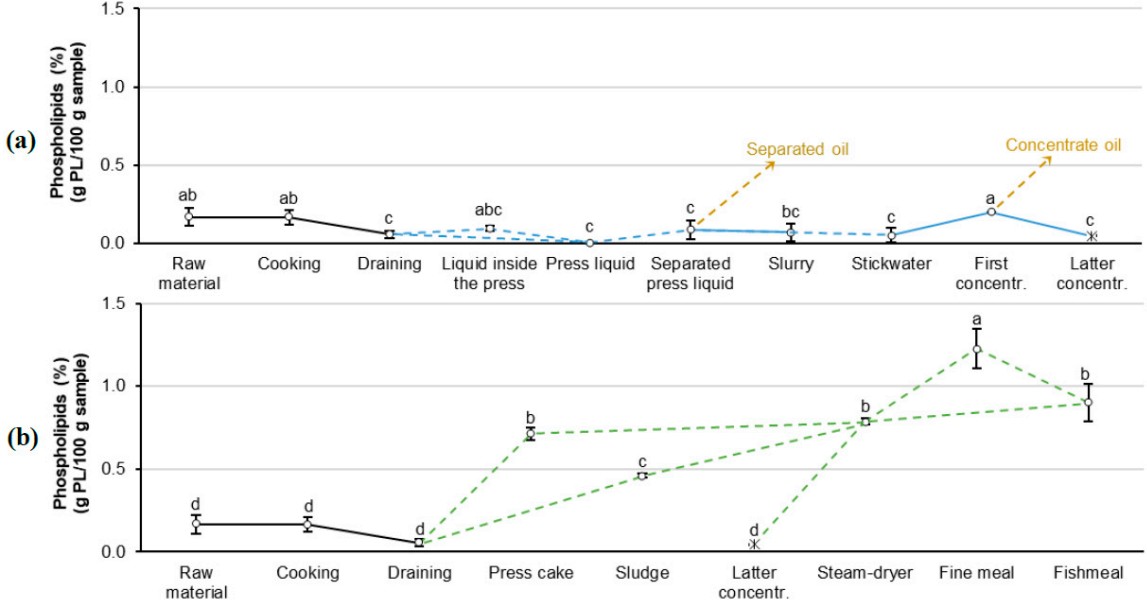

**Figure 5.** Phospholipid content in liquid streams (**a**) and solid streams (**b**) from the traditional fishmeal and fish oil processing line in Figure 1. Liquid streams are identified by a blue color, solid streams by a green color (**b**), and oil streams by a yellow color. A dashed line indicates where the process breaks up into multiple streams or where they join each other again. Solid lines indicate only one possible gateway. Letters indicate significant differences where $p < 0.05$.

When analyzing the PL content in the solid stream (Figure 5a), a clear increasing trend in PL content was observed after pressing and after the two drying steps. The trend was in agreement with the water removal and potential heat-induced lipid denaturation occurring during these processing

steps, increasing the relative phospholipid concentrations in the fishmeal [35,41]. Interestingly, the PL content of the fine meal in the air-drier was higher than the PL content of the final fishmeal, although the fine meal had a higher water content than the final fishmeal. The PL content changes during drying showed on a lipid basis that the fine meal had a significantly higher PL content (8.9 ± 0.1 g PL/100 g lipid) compared to the final meal (6.2 ± 0.8 g PL/100 g lipid). Since the same trend was seen both on a sample and a lipid basis, this difference cannot be explained by the changes in water content alone, but indicates a difference in the lipid composition of the two meal types. Moreover, the finer particle size of the fine meal would result in a proportionally higher surface area to volume ratio (A/V ratio) of the particles. This increased A/V ratio could increase the availability of microorganisms, oxidizing agents and other degrading factors to the components of the fine meal, compared with the final fishmeal. Hence, the question arises whether the fine meal should be mixed with the final fishmeal or not. Further processing of the solid streams individually (the press cake, sludge and the latter concentrate) might be beneficial to achieve better control of the characteristics of the final products, which in turn would result in both lower energy use and higher quality for each product.

3.1.5. Fatty Acid Composition (FAC) during Standard Processing (90 °C)

The fatty acid composition (FAC) of the samples was analyzed to give an overview of any compositional changes in the lipids during processing (Table 1). The overall FAC profile during processing was generally dominated by monounsaturated fatty acids (MUFA, 38.3–53.7 g/100 g lipid), followed by polyunsaturated fatty acids (PUFA, 13.3–34.6 g/100 g lipid), while containing a fairly low concentration of saturated fatty acids (SFA, 21.4–30.9 g/100 g lipid). The FAC of pelagic fish is highly dependent on the season and the place of the catch [30]. Lower levels of MUFA have been reported in mackerel (32.9 ± 1.4 g MUFA/100 g lipid) [30] compared with the raw material in the current study (42.2 ± 2.6 g MUFA/100 g lipid), while higher values of MUFA have been reported in herring (51.9 ± 0.4 g MUFA/100 g lipid) [32] than in the current study. The same trend was observed with SFA. As the raw material consisted of both herring and mackerel, these results could be expected. However, PUFA values of the raw material in the current study (31.4 ± 2.5 g PUFA/100 g lipid) are reported closer to mackerel (33.8 ± 0.8 g PUFA/100 g lipid) [30] than to herring (22.6 ± 0.6 g PUFA/100 g lipid) [32].

The solid streams entering the drying steps—the press cake, the sludge and the concentrate—differed significantly in FAC. The stickwater and the concentrate shared a similar FAC, which is not surprising as the stickwater is the precursor of the concentrate in the fishmeal processing line. The differences in FFA and PL in the fine meal and fishmeal could not be explained by differences in their fatty acid compositions. However, processing may change the structures or forms of the lipid molecules, as proteins associated with membranes are likely to be affected or influenced in their lipid environment [42], such as changing from the bilayer to the more stable micellar form [26]. Such structural changes could explain the availability of the FFA and PL for analysis in the fine meal compared with the final fishmeal. Such structural changes would, on the other hand, not influence the FAC as such, explaining the similar fatty acids content of the two fishmeal samples.

Both the final fishmeal and oil had PUFA and MUFA contents over 74 g/100 g lipid, including 23–28 g/100 lipid $n-3$ PUFA and a beneficial $n-3/n-6$ ratio of 3.2 ± 0.0 and 4.9 ± 0.1 in the fish oil and fishmeal, respectively. These fatty acid profiles are beneficial for various health effects and may decrease the risk of cardiovascular and coronary vascular diseases [43], prevent the development of breast cancer [44] and prostate cancers [45] and prevent obesity [46]. Although these fatty acid profiles are attractive, the high MUFA and PUFA content makes the products highly susceptible to lipid oxidation [47]. Increased lipid oxidation in food could, in turn, cause toxicity of the lipids as well as the secondary products, as the secondary products are more toxic than hydroperoxides [48].

**Table 1.** Fatty acid composition of samples from the fishmeal and fish oil processing line presented in Figure 1, with 90 °C temperature in the cooker. Results are presented as g fatty acid/100 g lipid as mean ± SD ($n = 3$) and are compared vertically within each column.

| Mackerel and Herring Blend at 90 °C | Lipid Content | SFA | MUFA | PUFA | EPA | EPA/DHA | $n-3$ PUFA | $n-3/n-6$ |
|---|---|---|---|---|---|---|---|---|
| **Raw material** | **19.5 ± 2.0 [c]** | **23.0 ± 0.5 [cde]** | **42.2 ± 2.55 [f]** | **31.4 ± 2.5 [abc]** | *8.7 ± 0.4 [a]* | **0.75 ± 0.0 [abc]** | **23.6 ± 1.7 [c]** | **3.3 ± 0.2 [ef]** |
| Pre-heating | 14.6 ± 0.6 [cd] | 23.1 ± 0.4 [cde] | 42.4 ± 1.0 [f] | 31.6 ± 0.5 [abc] | 8.4 ± 0.6 [a] | 0.7 ± 0.0 [cd] | 23.9 ± 0.7 [bc] | 3.5 ± 0.2 [cdef] |
| Cooking | 14.7 ± 0.7 [cd] | 22.4 ± 0.3 [cde] | 42.2 ± 0.7 [f] | 31.8 ± 0.6 [abc] | 8.3 ± 0.1 [ab] | 0.7 ± 0.0 [d] | 23.9 ± 0.5 [bc] | 3.4 ± 0.0 [def] |
| Draining | 11.4 ± 0.5 [de] | 21.5 ± 0.1 [de] | 46.7 ± 0.4 [cd] | 28.9 ± 0.3 [c] | 7.6 ± 0.1 [ab] | 0.7 ± 0.0 [d] | 21.9 ± 0.2 [c] | 3.4 ± 0.0 [def] |
| Liquid inside the press | 35.2 ± 8.6 [b] | 22.6 ± 0.1 [cde] | 43.2 ± 0.4 [ef] | 30.2 ± 0.4 [bc] | 7.6 ± 0.1 [ab] | 0.6 ± 0.0 [ef] | 23.6 ± 0.25 [c] | 4.2 ± 0.0 [b] |
| Separated press liquid | 15.5 ± 0.9 [cd] | 22.0 ± 0.2 [cde] | 44.3 ± 0.4 [def] | 30.1 ± 0.5 [bc] | 8.2 ± 0.2 [ab] | 0.8 ± 0.0 [ab] | 22.4 ± 0.4 [c] | 3.2 ± 0.0 [f] |
| Press cake | 8.8 ± 0.6 [def] | 28.2 ± 1.8 [b] | 49.0 ± 1.1 [bc] | 19.5 ± 3.1 [d] | 4.6 ± 0.8 [c] | 0.6 ± 0.0 [efg] | 14.9 ± 2.7 [d] | 3.8 ± 0.2 [bcd] |
| Sludge | 4.7 ± 0.2 [ef] | 30.9 ± 0.3 [a] | 53.7 ± 0.4 [a] | 13.3 ± 0.7 [e] | 2.8 ± 0.2 [d] | 0.6 ± 0.0 [fg] | 9.6 ± 0.6 [e] | 3.3 ± 0.3 [f] |
| Stickwater | 2.1 ± 1.0 [f] | 23.8 ± 0.0 [c] | 45.1 ± 0.9 [def] | 28.3 ± 1.7 [c] | 7.1 ± 0.7 [b] | 0.6 ± 0.0 [ef] | 22.0 ± 1.5 [c] | 3.8 ± 0.1 [bcde] |
| Latter concentrate | 3.9 ± 0.1 [f] | 21.4 ± 0.1 [e] | 45.6 ± 0.1 [de] | 28.9 ± 0.2 [c] | 7.7 ± 0.0 [ab] | 0.7 ± 0.0 [bcd] | 21.6 ± 0.2 [c] | 3.4 ± 0.0 [def] |
| Steam-dryer | 8.9 ± 0.2 [def] | 28.0 ± 1.2 [b] | 50.3 ± 2.0 [b] | 19.3 ± 2.9 [d] | 4.6 ± 0.8 [c] | 0.6 ± 0.0 [e] | 16.8 ± 2.9 [d] | 3.9 ± 0.3 [bc] |
| Fine meal | 12.3 ± 0.8 [d] | 23.4 ± 0.2 [c] | 38.8 ± 0.7 [g] | 34.1 ± 0.6 [ab] | 8.4 ± 0.1 [a] | 0.5 ± 0.0 [g] | 27.6 ± 0.6 [ab] | 4.8 ± 0.1 [a] |
| **Fishmeal** | **14.3 ± 0.2 [cd]** | **23.3 ± 0.3 [cd]** | **38.3 ± 0.6 [g]** | **34.6 ± 0.7 [a]** | **8.6 ± 0.1 [a]** | **0.5 ± 0.0 [g]** | **28.0 ± 0.7 [a]** | **4.9 ± 0.1 [a]** |
| **Final oil** | **99.7 ± 0.1 [a]** | **22.0 ± 0.1 [cde]** | **43.7 ± 0.2 [def]** | **31.3 ± 0.4 [abc]** | **8.6 ± 0.1 [a]** | **0.8 ± 0.0 [a]** | **23.2 ± 0.3 [c]** | **3.2 ± 0.0 [f]** |

Abbreviations: SFA (saturated fatty acids), MUFA (monounsaturated fatty acids), PUFA (polyunsaturated fatty acids), EPA (eicosapentaenoic acid), DHA (docosahexaenoic acid), $n-3$ PUFA (omega-3 PUFA) and $n-3/n-6$ (ratio between omega-3 and omega-6 fatty acids). [a–i] Letter indicates a significant difference between vertical results, where $p < 0.05$.

### 3.2. Effect of Different Cooking Temperatures

The analysis of the standard 90 °C process indicated that the current fishmeal processing line requires optimization, as shown in earlier sections. As a result of the inefficient breakdown of the raw material during the initial steps of the processing line, the final fishmeal was too high in lipid content. As muscle denaturation and degradation is highly dependent on the heat treatment chosen [14], it can be suggested that optimizations of the temperature in the cooker might obtain better separation between the lipid content and dry matter. Moreover, optimal cooking conditions have been questioned, where a minimum of 20 min above 70 °C for wild fish has been recommended [49], or 20 min at 75 °C [1] for optimal results, questioning the 95–100 °C cooking temperatures recommended by the FAO [5]. However, these temperatures aim primarily for the inactivation of parasites, viruses and bacteria and do not necessarily take separation of lipids and proteins into the equation. The second objective of the study was therefore to investigate the effect of different cooking temperatures (85 °C, 90 °C and 95 °C) on the water, lipid, FFA and PL composition on chosen sampling points throughout processing (Figure 6).

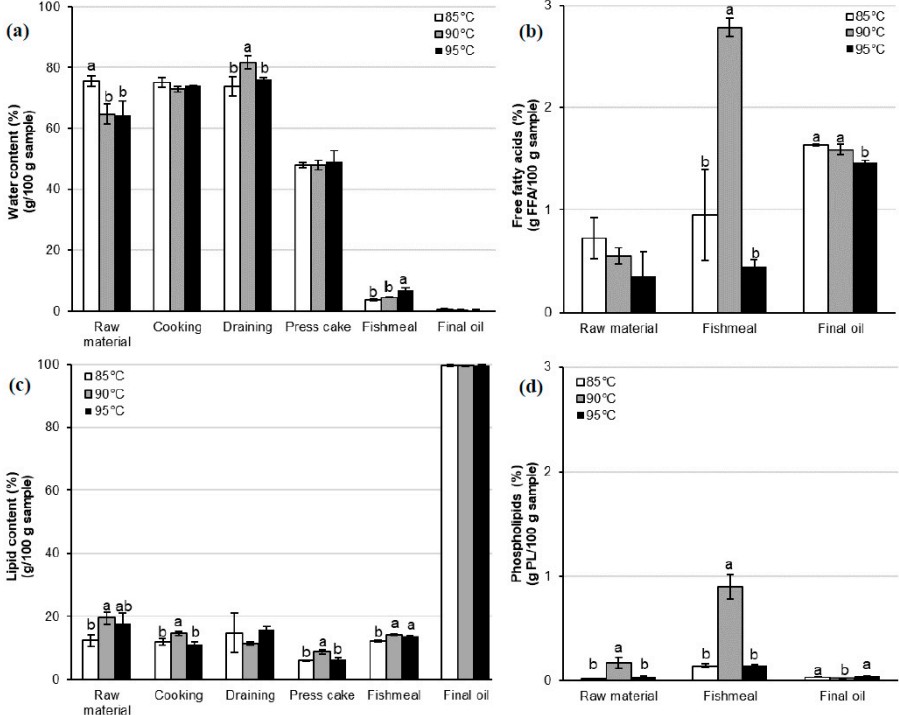

**Figure 6.** Measurements of water content (**a**); free fatty acids (**b**); lipid content (**c**); and phospholipids (**d**) from a traditional fishmeal and fish oil production line, presented in Figure 1 with a different temperature applied in the cooker (85 °C, 90 °C and 95 °C). All data is presented as a g/100 g sample as mean ± SD (*n* = 3).

Variations in the chemical composition of the raw material were observed between the three temperature runs, indicating that the raw material was highly heterogeneous. However, no significant differences were observed in the water content, and the variations in the lipid content decreased after cooking. After draining, the samples that underwent the 90 °C cooking were significantly higher in water content, although this was not reflected through the rest of the process. No systematic changes were seen in the water, lipid or FFA content through processing in relation to the observed variation in the chemical content of the raw material, indicating that any observed changes were indeed an effect of the processing treatments.

The water content of the fishmeal samples cooked at 85 °C and 90 °C were significantly lower than the fishmeal treated with 95 °C cooking. Furthermore, the fishmeal cooked at 85 °C was lower

in lipid content compared with the fishmeal samples treated at the other temperatures. Since low water and lipid content is beneficial for the stability of fishmeal, a processing temperature of 85 °C is recommended. No significant differences were observed in the water or lipid composition of the final oil between the three heat treatments.

When looking further at FFA and PL, the FFA concentrations were significantly higher in the fishmeal heated to 90 °C, while no significant differences in FFA and PL were seen in the final fishmeal or the fish oil at 85 °C and 95 °C treatments. Although slightly lower FFA were observed in the final oil at 95 °C, compared to the other heat treatments, this difference is too small to justify a recommendation of applying 95 °C heating.

Overall, the best results were obtained by lowering the temperature to 85 °C, resulting in a fishmeal of low water and lipid content, as well as low FFA and PL content. Lower PL content indicates a more efficient breakdown of the raw material. Higher temperature treatments are likely to denature proteins to a greater extent [13], decreasing their quality and, therefore, also their application possibilities for human consumption. Analysis of the protein quality changes during processing is, however, a matter for a later study.

## 4. Conclusions

Analysis of the Atlantic mackerel raw material indicated that, although up to three days had passed from catch to processing, the raw material was at a good lipid quality. Large variations in raw material characteristics may, though, make processing problematic and less homogeneous, and long delays between catch and processing may increase such raw material quality variations. It might, therefore, be beneficial for the processing companies to shorten any processing delays to open the possibility of producing higher quality fishmeal and fish oil products. Currently, several companies own trawlers that process the fishmeal onboard directly from catching [1], which could eliminate the processing delay.

One of the main problems of pelagic fishmeal production lies in the high lipid content of the raw material and problems in lipid removal from the fishmeal. Detailed analysis of the chemical changes during processing revealed that the solid streams entering drying have different chemical compositions. Hence, different processing is suggested depending on the characteristics of each stream, such as different drying times for the press cake (50% water) and the latter concentrate (80% water). Moreover, all the solid streams entering the dryers were too high in lipid content, meaning that the initial breakdown of the raw material was not sufficient. Furthermore, the standard process at 90 °C revealed poor effectiveness of water removal during draining, as well as an increase of FFA and PL during the steam and air drying steps of the fishmeal, indicating that the process required optimization.

During the analysis of different cooking temperatures in the mackerel and herring blend, it was evident that the cooking steps had a highly important role in the lipid removal from the fishmeal processing. By lowering the temperature in the heater to 85 °C, the water and lipid content of the fishmeal was lowered, as well as contributing to lower the FFA and PL values, indicating the production of a more stable product at 85 °C compared with the standard 90 °C. Moreover, the PL values were lower at 85 °C, indicating a more efficient breakdown of the raw material. In addition to a higher quality fishmeal, energy costs can be decreased, as fishmeal and fish oil factories are operating with a cooker at temperatures up to 95–100 °C. Moreover, performing a life cycle assessment (LCA) is suggested to investigate the environmental impact of the processing. Lowering the temperature in the heater to 85 °C can therefore be recommended.

Further recommendations include investigation of ways to break down the raw material more efficiently during the first steps in the production line, which could be applied in commercial fishmeal and fish oil factories. As the diversity of the fish protein is high, as well as the volume, a possible solution for homogenizing the raw material is applying enzymatic technology, but fish protein hydrolysates are currently being produced industrially [50]. Drying affected the FFA concentrations and, hence,

optimizing the drying of the different solid streams is recommended to receive the highest value possible and open up the possibility of producing products intended for human consumption.

**Author Contributions:** Conceptualization, G.S.H., Ó.O., S.A., and M.G.; methodology, G.S.H., S.A., and M.G.; software, G.S.H.; validation, G.S.H.; formal analysis, G.S.H.; investigation, G.S.H; resources, S.A. and M.G.; data curation, G.S.H.; writing—original draft preparation, G.S.H.; writing—review and editing, G.S.H., Ó.O., S.A., and M.G.; visualization, G.S.H., Ó.O., S.A., and M.G.; supervision, Ó.O., S.A., and M.G.; project administration, S.A. and M.G.; funding acquisition S.A. and M.G. All authors have read and agreed to the published version of the manuscript.

**Funding:** This research was funded by the AVS (The Added Value of Seafood) fund of the Ministry of Fisheries and Agriculture in Iceland (grant number: R18 031-18), the Rannís Technology Development Fund (no. 198883-0611), and the University of Iceland research fund.

**Acknowledgments:** The work was carried out at the University of Iceland and Matís ohf. The authors thank Síldarvinnslan hf. for access to their facilities, assistance, and raw materials.

**Conflicts of Interest:** The authors declare no conflict of interest.

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
