# Peer review of "The Effects of Varying Heat Treatments on Lipid Composition during Pelagic Fishmeal Production"

_processes, doi:10.3390/pr8091142_

Round 1
Reviewer 1 Report
- Line 43 - 46: Point to denaturation of muscle of lean fish (cod) at 30C. Are the results from cod-muscle directly applicable to pelagic species by-products with a higher lipid and connective tisse conent? Additional references of studies from studies of protein in general could perhaps underline the statement in line 45-46.
- Line 80: Please clearify the purpose of the pre-heating step (ease of transportation/pumping?).
- Line 133-135: Please explain better why there was an increase in the water content due to heating. Was the initial, raw-material measurments not reprentative beceause of heterogenous mixture? Do heating increase absorbion of water?
- Line 194-196: a relatively low FFA content was observed...Relatively to what? (refenrce to other relevant FFA studies would be helpfull).
- Line 209: See comment above
- Line 220: Consider to change to: the raw-materials exposure (passive) to oxygen rather than using "access" (active).
- Line 350-51: A reference/comparison to fishmeal industry that produce human grade fish-meal from fresh by-products could be usefull. Eg salmon fishmeal and whitefish meal produced at-sea on fabric trawlers.
- Line 367: A few words regarding the savings in energy by lowering the temperature 10C is should be mentioned as a point of interest to fishmeal producers worldwide.
- Line 369: Mention enzyme technology as a possible solution.
Author Response
"Please see the attachment."

Reviewer 2 Report
To the Authors,
in the manuscript (Processes-922760), titled "The effects of varying heat treatments during pelagic fishmeal production" by Hilmarsdottir et al. studied the " lipid quality of pelagic fishmeal and fish oil processing of mackerel and herring cut-offs, and the effect of temperature changes in the cooker" during production. "Results showed that the standard procedures at 90°C included ineffective draining- and concentration steps".
It is an interesting study with important real-life application. Fish oil and fishmeal production are current important topics of research because of the issues raised in terms of quality characteristics and safety for human consumption and environmental sustainability (origin of raw materials, cleaner production incl. wastewater/residues production and treatment). Authors had the opportunity of carrying out the "trials"/"runs" of an experiment in an industrial setting. This context suggests the use of more "appropriate" (stats) experimental designs such as the 2-k factorial and central composite designs (see e.g. Anderson, M.J., Whitcomb, P.J., 2007. DOE Simplified, Third. ed. CRC Press Inc., Boca Raton, Florida, USA, Myers, R.H., Montgomery, D.C., Anderson-Cook, C.M., 2016. Response Surface Methodology: Process and Product Optimization Using Designed Experiments, 4th ed. Wiley, Hoboken, NJ, USA, Anderson, M.J., Whitcomb, P.J., 2005. RSM Simplified: Optimizing Processes Using Response Surface Methods for Design of Experiments. Productivity Press, New York, USA.). Authors are encouraged to complement the contextualization of their study in the Introduction to emphasize its scientific and practical relevance.
A few typos are noted and some suggestions are made to the text as the paper reads well. Some assertions require citation(s) or further completion/clarification. Figures could be improved in terms of data visualization criteria.

Author Response
"Please see the attachment."

Reviewer 3 Report
General remarks:
This article is well written and has very interesting results that can make a significant contribution for the study of fishmeal production from pelagic fishes.
The article follows well the “Instructions for Authors” and is well structured and very well written.
In order to facilitate the location of my comments, I am going to present them organized by line number of the “processes-922760-peer-review-v1” pdf file.
Revision comments:
Line 2: The title is two generic and can mislead the reader. The article focus only on lipid composition, and that issue should be put on the tile.
Suggestion: The effects of varying heat treatments on lipid composition during pelagic fishmeal production
Line 51: Zooplankton species name should be in italics.
Line 59: Change “non-heterogeneous” to “heterogeneous” or “non-homogeneous”.
Figure 1: In the boxes “Pre-heating” and “Cooking” delete the minus signal before the 20 minutes (a period of time cannot be negative)
Figure 2: I understand the aim of putting all the results in just one figure, but the output is too complex and should be divided in four different figures, that should be put alongside the text that cites them.
Line 133 – Convert 2A-2B in one figure and put it around here
Line 166 – Convert 2C-2D in one figure and put it around here
Line 209 – Convert 2E-2F in one figure and put it around here
Line 242 – Convert 2G-2H in one figure and put it around here
Line 232: Genus Alcaligenes should be in italics.
Author Response
"Please see the attachment."
